# Post-Dilatation of New-Generation Self-Expandable Transcatheter Aortic Valves Does Not Increase Atrioventricular Conduction Abnormalities

**DOI:** 10.3390/diagnostics13030427

**Published:** 2023-01-24

**Authors:** Grégoire Massoullié, Nicolas Combaret, Géraud Souteyrand, Jean Pascal Salazard, Bruno Pereira, Frédéric Jean, Pascal Motreff, Ouarda Taghli-Lamallem, Guillaume Clerfond, Romain Eschalier

**Affiliations:** 1Cardiology Department, Clermont-Ferrand University Hospital, 63000 Clermont-Ferrand, France; 2Institut Pascal, SIGMA Clermont, Centre National de la Recherche Scientifique, Université Clermont Auvergne, Clermont-Ferrand University Hospital, 63000 Clermont-Ferrand, France; 3Biostatistics Unit (Clinical Research and Innovation Direction), Clermont-Ferrand University Hospital, 63000 Clermont-Ferrand, France

**Keywords:** post-dilatation, conductive disorders, TAVR, PPM

## Abstract

The impact that post-dilatation has on the risk of experiencing conduction disorders after post-transcatheter aortic valve replacement with self-expanding valves (SE-TAVR) is unclear. We compared the rate of developing an atrioventricular (AV) high-grade conduction disorder and permanent pacemaker implantation (PPI) in post-TAVR patients undergoing post-dilatation. We enrolled patients with severe symptomatic calcified aortic stenosis (CAS) who were undergoing SE-TAVR between 1 January 2016, and 19 April 2019 at a single French center. Of the 532 patients treated with SE-TAVR, 417 subjects (78.4%) received Corevalve Evolute R and 115 subjects (21.6%) received the latest-generation Corevalve Evolute Pro valve. In total, 104/532 patients (19.5%; 21.6% with Evolute R vs. 12.2% with Evolute Pro, *p* = 0.024) required post-dilatation. Evolut R was associated with an increased risk of post-dilatation (odds ratio 2.1 (1.01–4.33, *p* = 0.046)). We did not observe any post-dilatation increases in AV or in intra- and interventricular conduction disorders. In total, 26.1% of participants needed PPI within the first 30 post-procedure days (*p* = 0.449). Post-dilatation was not associated with a higher PPI risk (subdistribution hazard ratio 1.033 (0.726–1.471); *p* = 0.857). No significant differences existed between the groups in terms of one-year mortality (10.3%; *p* = 0.507). Post-dilatation in SE-TAVR did not increase the rate of electrical conduction disorders and PPI in the early implantation phase. The latest generation of SE-TAVR valves was associated with less need for post-dilatation.

## 1. Introduction

Transcatheter aortic valve replacement (TAVR) is a viable option for all patients with severe symptomatic aortic stenosis [1,2,3,4,5,6]. The first-, second-, and third-generation (SE-TAVR) self-expanding valves Corevalve™, Corevalve™ Evolut R, and Corevalve™ and Evolut Pro (all Medtronic Inc), respectively, ameliorate periprocedural outcomes (a reduced aortic gradient and reduction in morbimortality) and decrease the risk of stroke, aorta disruption, and vascular complications [7,8]. Since the discontinuation of systematic pre-dilatation, the post-dilatation rate of percutaneous valves has increased and coincides with the marketing of the latest generation of valves [9].

No specific guideline exists regarding the balloon post-dilatation that is usually performed to reduce periprosthetic leakage or prosthesis underdeployment, which is associated with poor outcomes [10,11]. Post-dilatation optimizes the apposition of the valve frame on the annulus, but the procedure causes a mechanical stress to the aortic ring, the left ventricular chamber, and the newly implanted valve [12,13]. This could increase the risk of conduction disturbances, which are common complications with first-generation self-expanding valves (25% to 28%), and could alter the motion of the prosthetic valve [13,14,15,16,17]. It is unclear whether a relevant risk for these complications exists with the newer-generation aortic valve prostheses Corevalve Evolut R and Evolut Pro.

The objective of the present study was to evaluate the post-dilatation rate in TAVR procedures using second- and third-generation self-expandable valves and to evaluate their impact on the occurrence of conduction disturbances and pacemaker implantation (PPI) after TAVR.

## 2. Materials and Methods

### 2.1. Study Population

This was a retrospective single-center analysis of patients enrolled in the France-TAVI registry who underwent a femoral TAVR using second- and third-generation SE-TAVR Corevalve™ from January 2016 to May 2019 at the university-affiliated hospital of Clermont-Ferrand, France. Patients were excluded from the analysis if they had a valve-in-valve procedure. All patients provided written, informed consent [18]. The institutional review board of the French Ministry of Higher Education and Research (CCTIRS) and the National Commission for Data Protection and Liberties (CNIL) approved the France-TAVI registry.

### 2.2. TAVR Procedure and Data Collection

The indication for TAVR, the type of transcatheter valve, and access were established by the multidisciplinary site heart team. Standard implant procedures were used [19]. Clinical outcomes were defined according to the Valve Academic Research Consortium 2 (VARC 2) definitions [20].

Coronary angiography was performed systematically, and patients were treated according to the identified angiographic lesion. A pre-procedural cardiac computed tomography scan (CT-scan) was performed to evaluate perimeter-based annular diameters for valve size selection.

All relevant information regarding patients’ characteristics and procedures using the second and third generations of Corevalve and clinical records were collected prospectively. Patient data were reviewed from procedural records and electronic medical records.

Upon TAVR completion, two interventional cardiologists jointly made the post-dilatation decision in the case of: 1) the underdeployment of the prosthesis, based on two perpendicular X-ray analyses; and/or 2) paravalvular leakage grades ≥2, according to the classification of Sellers by angiography, or an alteration in the hemodynamic performance with an aortic regurgitation index less than 25% [21,22].

### 2.3. Outcomes: Electrical Conduction Disorders, Clinical Follow-Up

The electrocardiogram (ECG) was monitored before TAVR, post-procedure for 3 days (day 0 [D0] to D3), and at the day of hospital discharge (between day 5 and day 7).

The timing and type of electrical cardiac disorders have been described. Requirement of permanent pacemaker implantation after TAVR is indicated in the case of a third-degree AV block, second-degree type II AV block, left bundle branch block (LBBB) associated with right bundle branch block (RBBB), bradycardia with atrial fibrillation (bradyarrhythmia), and complete LBBB block associated with increased H-V (His bundle–ventricular activation) >70 ms (LBBB-TAVR study) [23].

The occurrence of conduction disturbances or PPI during index hospitalization for TAVR was the primary endpoint. Death within 30 days or 1 year after TAVR was the secondary endpoint.

### 2.4. Statistical Analysis

Data are presented as numbers and percentages for categorical variables and as means ± standard deviation for continuous variables. Comparisons between groups (post-dilatation vs. control) and analysis of outcomes were performed using the Chi square test or Fisher’s exact test, as appropriate for categorical variables, and using the Student’s *t*-test (Mann–Whitney U test for non-normally distributed data) for continuous variables. Univariate analyses were conducted to identify variables with a relationship to the primary outcome. If the *p* value was less than 0.10, variables were selected for multivariable analysis. Covariates for multivariable analysis were chosen by taking into account the univariate analysis, potential clinical relevance, and limited sample size [24]. The proportional-hazards hypothesis was verified using Schoenfeld’s test and plotting residuals. The results are presented as subdistribution hazard ratios (SHR) and 95% confidence intervals (CI). Normality was assessed graphically using the Shapiro–Wilk test. All tests were two-sided, with a *p*-value < 5% considered statistically significant. Statistical analyses were carried out using Stata v14 (StataCorp, College Station, TX, USA).

## 3. Results

### 3.1. Baseline Characteristics

From January 2016 through May 2019, a total of 933 patients underwent TAVR. Second- and third-generation Corevalves were used in 532 subjects, which were included in the final analysis. Post-dilatation was performed in 104 SE-TAVR patients (19.5% of the analyzed population) (Table 1, Figure 1).

At baseline, patients who underwent post-dilatation had a lower body mass index (*p* = 0.001), a more frequently altered renal function (*p* = 0.032), and a more severe aortic stenosis (lower aortic valve area and higher mean gradient; *p* < 0.001 for both). We observed no other differences, including in AV and in inter- and intra-ventricular conduction disorders or between patients undergoing and not undergoing post-dilatation (Table 1).

Most patients (410; 78.4%) had a sinus rhythm, 154 (29.4%) had a type 1 AV block (mean PR interval: 192 ± 38 ms), 26 (5.0%) had an LBBB, and 64 (12.2%) had an RBBB (Table 2).

### 3.2. Procedural Characteristics (Appendix A)

Balloon aortic pre-dilatation was performed in 39 patients, 33 of whom received a TAVR completed before 2017. Post-dilatation (*n =* 104) was performed after TAVR implantation for paravalvular leakage ≥ II (*n =* 68) or prosthesis malapposition (*n =* 36). The size of the prosthesis was not associated with post-dilatation (*p* = 0.555). A prosthesis oversize rate of ≤10% was similar between both non-dilated and dilated groups (7.9% vs. 12.5%, *p* = 0.142). We observed post-dilatation significantly less often in the Corevalve Evolut Pro group (Evolut R vs. Evolut Pro: 21.6 % vs. 12.1 %, *p* = 0.024). Before post-dilatation, 77 patients had a moderate or severe regurgitation (nine due to malapposition): 66 (85.7%) patients with a second-generation Corevalve and 11 (14.3%) patients with a third-generation Corevalve. In total, 54 (51.4%) and 23 (22.1%) patients experienced moderate and severe regurgitation, with no difference between the valve types (*p* = 0.532). After post-dilatation, 36 patients (34.6%) had a moderate leak and 3 (2.9%) a severe leak. No significant difference according to the valve type existed (see Appendix A, *p* = 0.378).

No increase in AV or ventricular conduction abnormalities occurred during the TAVR procedure.

In the multivariate analysis, Corevalve Evolut R implantation was an independent predictor of post-dilatation SHR: 2.1 (1.01–4.33, *p* = 0.046).

### 3.3. Conduction Abnormalities and Permanent Pacemaker Implantation (Table 2 and Table 3, Figure 2)

At day 1 after TAVR, the mean PR interval increased by 11 ± 29 ms and 10 ± 27 ms in the dilated and nondilated groups, respectively (*p* = 0.878), and the mean QRS duration increased by 25 ± 26 ms and 27 ± 25 ms, respectively (*p* = 0.415). The mean QRS interval was 123 ± 30 ms with an overall LBBB rate of 35.7% (184 patients, 120 with de novo LBBB) and with no differences between the dilated and nondilated groups.

**Table 3 diagnostics-13-00427-t003:** Electrocardiogram description and electrical conduction abnormalities.

	Total (*n =* 532)	No Post-Dilatation Group (*n =* 428)	Post-Dilatation Group (*n =* 104)	*p*-Value
PR interval ^#^ (ms)				
Before TAVR	191 ± 38	191 ± 37	195 ± 40	0.456
Day 1	201 ± 41	200 ± 40	205 ± 44	0.366
Discharge	212 ± 48	212 ± 47	211 ± 50	0.842
QRS length ^#^ (ms)				
Before TAVR	100 ± 25	100 ± 25	97 ± 23	0.301
Day 1	123 ± 30	123 ± 27	123 ± 31	0.986
Discharge	121 ± 30	121 ± 30	120 ± 26	0.704
New permanent pacemaker implantation	149 (28.0%)	123 (28.7%)	26 (25.0%)	0.446
Etiologies of pacemaker implantation				0.294
Sinus bradycardia	5 (3.4%)	3 (2.4%)	2 (7.7%)	
Complete AV block or Mobitz 2 AV block	107 (71.8%)	87 (70.7%)	20 (76.9%)	
LBBB and HV ≥ 70 ms	30 (20.1%)	26 (21.1%)	4 (15.4%)	
Other abnormalities	7 (4.7%)	7 (5.7%)	0 (0.0%)	
Timing of conduction abnormalities				0.468
During procedure	53 (35.6%)	44 (37.7%)	9 (34.6%)	
≤24 h	35 (23.5%)	14 (25.2%)	4 (15.4%)	
>24 h	61 (41.0%)	48 (39.0%)	13 (50.0%)	

AV: atrioventricular; HV: His bundle–ventricle length; LBBB: left bundle branch block. ^#^ Paced rhythm excluded.

At discharge, the results of ECG monitoring between post-implantation and discharge showed a 10 ± 38 ms increase in the mean PR interval and a 2 ± 21 ms increase in the QRS interval, without differences between the groups.

The LBBB rate at discharge was 35.6% (*n =* 179). Thirty-eight patients developed LBBB between day 1 and discharge, and 43 patients had a transient de novo LBBB.

A total of 139 patients (26.1%) required PPI within the first 30 days post-TAVR (*p* = 0.446). The median delay of PM implantation was 3 (2–6) days. The indications were a complete AV block in 58% (86), AV block type 2 in 5% (8), and alternating bundle branch block pattern in 8% (12) of patients. We did not observe an increase in the AV block (*p* = 0.839). The multivariate analysis results showed that post-dilatation was not associated with PPI (SHR = 1.033 (0.726–1.471), *p* = 0.857) (Figure 3).

**Figure 2 diagnostics-13-00427-f002:**
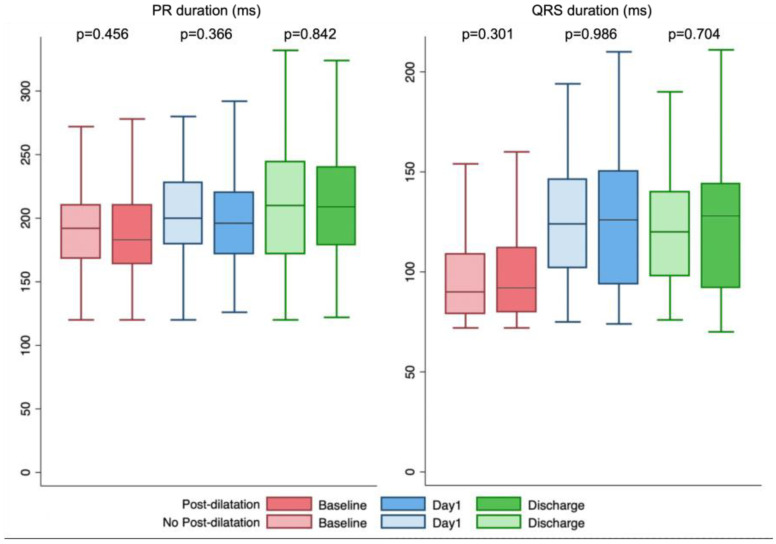
PR and QRS duration at baseline, Day 1 after TAVI and at discharge in patient with or without post-dilatation.

### 3.4. TAVR Clinical Outcomes (Appendix A)

No significant differences existed in the occurrence of complications during pre- and postprocedure between patients with and without post-dilatation. At 1 year, no difference in NYHA status existed (*n =* 80, 2.0 ± 0.6 vs. 2.0 ± 0.6, *p* = 0.405).

The all-cause mortalities at 30 days and 1 year were not significantly different between the groups (2.4%, *p* = 0.302 at 30 days and 10.3%, *p* = 0.507 at 1 year). Of the patients who died and received post-dilatation, six (75%) had moderate-to-severe leakage.

## 4. Discussion

To the best of our knowledge, this is the first study dedicated to evaluating new-generation SE-TAVR valves with a focus on the need for post-dilatation as well as on the occurrence of conduction abnormalities and permanent PPI. The main findings are that the latest SE-TAVR generation (i.e., Corevalve Evolut Pro) reduces the need for post-dilatation and that post-dilatation does not increase the risk of conduction disturbances and PPI after TAVR.

In an era where ambulatory TAVR is emerging, questioning the risk of high-grade conduction disorders is crucial, both in terms of reducing the risk of complications and in terms of the medico-economic approach. Several reports have described predictors of cardiac conduction disturbances leading to PPI [14]. The increased occurrence of conduction abnormalities depends on patient-dependent parameters (pre-existing RBBB, calcium score of the valve, left ventricular outflow tract width, etc.), the use of self-expandable valves, and procedural characteristics (predilatation, implant depth, etc.) [14,25]. These conduction abnormalities motivate a prolonged post-TAVR monitoring, preventing pacing or a risk stratification using electrophysiology. We herein demonstrated that the additional post-dilatation stress on the implanted valve was not one of these risks factor and did not modify PPI risk.

In the case of prosthesis underdeployment, post-dilatation is a required strategy to lessen periprosthetic leakage (12–20%), which is associated with increased mortality [26,27]. A few studies on first-generation valves with a short follow-up and a certain discrepancy have addressed post-dilatation outcomes with a specific focus on conduction abnormalities and post-TAVR PPI [15,17,28]. The post-dilatation of first-generation self-expandable valves and balloon-expandable valves are associated with a higher PPI rate, 43.9% vs. 7.0%, *p* < 0.001, without a distinction between valve types being present [29,30,31]. In the largest published cohort by Harrison et al., the authors performed a post-dilatation in 782 patients (22%) [17]. At 30 days, the PPI rate was similar in patients with and without post-dilatation (22.5% vs. 20.4%). A higher LBBB incidence after the post-dilatation of self-expandable valves (26.1% vs. 32.6%, *p* = 0.038) was described without a significant increase in PPI rates.

Despite the similar post-dilatation and post-TAVR PPI rate, comparing our population to the population of the aforementioned studies is difficult. First, the authors of the previous reports explored first-generation SE-TAVR (2005–2011, 2009–2011, and 2009–2013, respectively) [15,17,28]. Second, with an almost systematic predilatation (82.1%) [17], it is unclear whether the authors performed post-dilatation procedures on the same population that we did. Third, few ECG data allow us to understand the evolution of post-TAVR conduction disorders.

With the present study, we provide a deeper knowledge regarding the postprocedural impact of post-dilatation on the AV conduction from ECG. According to current practice, physicians only perform predilatation in 6% of TAVR cases, as no scholars have reported a considerable reduction in post-TAVR complications [9]. These results are encouraging with regard to the postprocedural safety of post-dilatation, and they should not encourage specific post-TAVR monitoring. From a practical point of view, post-dilatation might enhance the fit between the bioprosthesis and the aortic annulus without altering the conduction system, and it might further normalize the radial force applied by the valve deployment without increasing the risk of conduction disorders.

Using the last Corevalve Evolut Pro resulted in a lower post-dilatation rate (12.2% vs. 21.6%, *p* = 0.024). The new-generation SE-TAVR has a similar size and strut as the Evolut R valve, but it was designed with an outer pericardial wrap at the annular landing zone, increasing surface contact with the native valve and enhancing its annular sealing. This outer skirt achieves to reduce the severe paravalvular leakage and to lessen post-dilatation use, although the Corevalve Evolut Pro structure increases radial force and thus allows for a mechanical reduction in paravalvular leak rates and malposition. We did not observe that the aortic annulus size or oversizing were associated with the need for post-dilatation, a finding described for first-generation devices using 26, 29, and 31 mm valves [15]. The development of larger-diameter valves, including the 34 mm one, appears to have enhanced the aortic annular sealing and enabled the choice of a suitable device size.

### Study Limitations

This was a single-center study and may not be representative of the practice at other sites. However, the main objective of this study was to assess the clinical impact of postimplant balloon valvuloplasty in a dedicated large sample in actual clinical practice. We did not consider predictors such as implantation depth and the device landing zone calcification score. We based the decision to use post-dilatation on defined criteria and the agreement of two interventional cardiologists, both of whom had thorough experience performing TAVR for more than 5 years. Conducting a more thorough study on the link between anatomy, post-dilatation, and the appearance of conduction anomalies would be informative. We excluded balloon-expandable valves from our cohort. Of the 362 patients implanted, only 1 had received post-dilatation. In the most recent literature, patients undergoing the post-dilatation of an expandable balloon or self-expanding valve shared similar clinical characteristics, with no mechanical complications or impact in terms of 1-year mortality. No description of procedure electrocardiogram changes is available for patients undergoing the post-dilatation of a self-expanding valve.

## 5. Conclusions

Post-dilatation of second- and third-generation Corevalve™ self-expandable TAVR valves did not increase the risk of developing conduction disturbances or PPI. Using the latest-generation SE-TAVR considerably reduced periprosthetic regurgitation and reduced the need for post-dilatation.

## Figures and Tables

**Figure 1 diagnostics-13-00427-f001:**
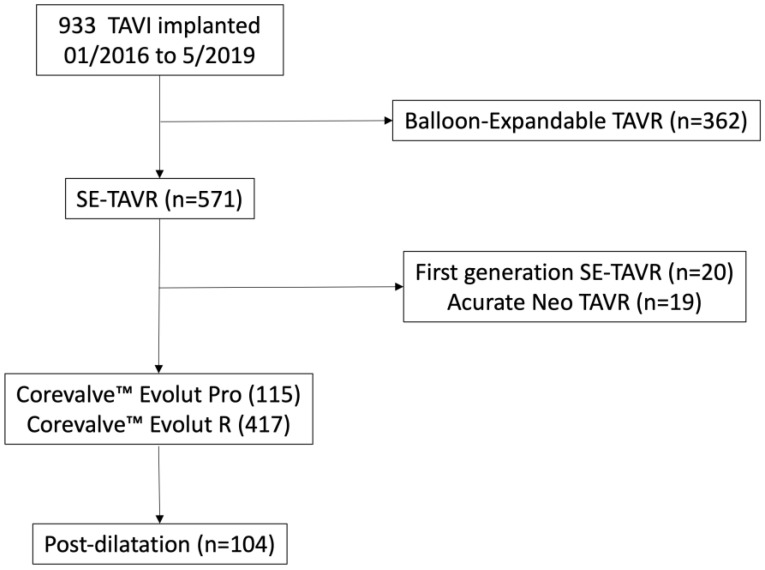
Study flowchart. SE-TAVR: self-expandable transaortic valve replacement.

**Figure 3 diagnostics-13-00427-f003:**
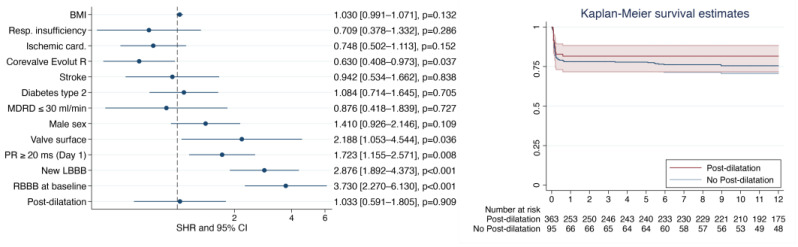
Multivariate analysis adjusted on mortality and previous pacemaker implantation (**left**), and Kaplan–Meier survival curve without a pacemaker implanted for high-grade conduction disorder (**right**). BMI: indexed body mass index; Resp. insufficiency: respiratory insufficiency; LBBB: left bundle branch block; RBBB: right bundle branch block.

**Table 1 diagnostics-13-00427-t001:** Baseline characteristics.

	Total (n = 532)	No Post-Dilatation Group (n = 428)	Post-Dilatation Group (n = 104)	p-Value
Age, yrs	82 ± 6	82 ± 6	83 ± 6	0.361
Female, no. (%)	260 (48.8%)	212 (49.5%)	48 (46.1%)	0.536
Logistic EuroSCORE	13.8 ± 8.2	13.9 ± 8.5	13.6 ± 7.3	0.717
Logistic EuroSCORE II	4.2 ± 3.0	4.2 ± 3.2	4.0 ± 2.1	0.528
BMI (kg/m^2^)	27.1 ± 5.2	27.3 ± 5.3	25.7 ± 4.2	0.001
COPD, no. (%)	65 (12.2%)	57 (13.3%)	8 (7.7%)	0.116
NYHA (Status)	2.3 ± 0.6	2.2 ± 0.6	2.3 ± 0.6	0.033
AF, no. (%)	114 (21.4%)	97 (22.6%)	17 (16.3%)	0.159
GFR median (ml/min/1.72 m^2^)	52.2 ± 19.7	52.7 ± 20.5	48.6 ± 16.9	0.032
Diabetes mellitus, no. (%)	161 (30.2%)	131 (30.6%)	30 (28.9%)	0.828
Previous PPI, no. (%)	45 (8.4%)	42 (9.8%)	3 (2.9%)	0.023
Stroke /TIA, no. (%)	44 (8.3%)	34 (7.9%)	10 (9.6%)	0.579
Prior cardiac surgery, no. (%)	48 (9.1%)	40 (9.3%)	8 (7.7%)	0.469
Prior PCI, no. (%)	205 (38.6 %)	170 (39.8%)	35 (34.0%)	0.405
Echocardiographic assessment				
LVEF (%)	60 ± 11	59 ± 11	60 ± 11	0.258
Aortic valve area (cm^2^)	0.73 ± 0.22	0.74 ± 0.22	0.64 ± 0.22	<0.001
Mean gradient (mmHg)	45 ± 15	43 ± 13	55 ± 18	<0.001
Moderate Aortic regurgitation, no. (%)	67 (12.6%)	54 (12.6%)	13 (12.5%)	0.974
Moderate Mitral regurgitation, no. (%)	88 (16.5%)	67 (15.5%)	21 (20.4%)	0.688
Aortic annular diameter (mm, CT)	23.8 ± 2.4	23.8 ± 2.4	23.8 ± 2.1	0.844

BMI: body mass index; COPD: chronic obstructive pulmonary disease; CT: cardiac computed tomography; LVEF: left ventricular ejection fraction; NYHA: New York Health Association; AF: atrial fibrillation; GFR: glomerular filtration rate; TIA: transient ischemic attack; PASP: pulmonary artery systolic pressure; PCI: percutaneous coronary intervention; PPI: permanent pacemaker implantation.

**Table 2 diagnostics-13-00427-t002:** ECG characteristics before and after TAVR and at discharge.

	ECG Pre-TAVI				ECG J1-TAVI				ECG Discharge-TAVI			
	Total (523)	No Post-Dilatation (423)	Post-Dilatation (100)	*p* Value	Total (515)	No Post-Dilatation (417)	Post-Dilatation (98)	*p* Value	Total (498)	No Post-Dilatation (401)	Post-Dilatation (97)	*p* Value
Sinus rhythm (%)	410 (78.4%)	328 (77.5%)	82 (82%)	0.631	371 (72%)	297 (71.2%)	74 (75.5%)	0.726	363 (72.9%)	290 (72.3%)	73 (75.3%)	0.632
Heart rate (bpm)	72 ± 14	72 ± 15	72 ± 14	0.647	77 ± 15	78 ± 16	74 ± 14	0.028	76 ± 14	76 ± 14	76 ± 15	0.897
PR intervals (ms)	192 ± 38	191 ± 37	195 ± 40	0.458	201 ± 41	200 ± 40	205 ± 44	0.367	212 ± 48	212 ± 47	211 ± 50	0.842
Type 1 AVB (%)	154 (29.4%)	122 (28.8%)	32 (32%)	0.85	179 (34.8%)	138 (33.1%)	41 (41.8%)	0.203	189 (38%)	150 (37.4%)	39 (40.2%)	0.878
QRS intervals (ms) *	100 ± 25	100 ± 25	97 ± 23	0.302	123 ± 30	123 ± 31	123 ± 27	0.985	121 ± 30	121 ± 30	120 ± 27	0.704
QTc (ms)	423 ± 31	424 ± 32	421 ± 28	0.433	463 ± 47	464 ± 47	460 ± 46	0.524	437 ± 40	438 ± 39	433 ± 41	0.29
Hemiblock												
HAFB (%)	83 (15.9%)	63 (14.9%)	20 (20%)	0.255	48 (9.3%)	40 (9.3%)	8 (1.6%)	0.598	39 (7.8%)	39 (9.7%)	8 (8.2%)	0.875
HPFB (%)	4 (0.8%)	4 (0.9%)	0 (0%)	0.322	9 (1.7%)	7 (1.7%)	2 (0.4%)	0.838	1 (0.2%)	1 (0.2%)	1 (1%)	0.042
QRS type												
Narrow QRS (%)												
RBBB (%)	26 (5%)	22 (5.2%)	4 (4%)	0.583	44 (8.5%)	38 (9.1%)	6 (6.1%)	0.302	27 (5.4%)	27 (6.7%)	5 (5.2%)	0.89
LBBB (%)	64 (12.2%)	52 (12.3%)	12 (12%)	0.864	184 (35.7%)	147 (35.3%)	37 (37.7%)	0.813	179 (35.9%)	179 (44.6%)	37 (38.1%)	0.642
NICD (%)	17 (3.3%)	16 (3.8%)	1 (1%)	0.149	12 (2.3%)	10 (2.4%)	2 (2.04%)	0.799	14 (2.8%)	14 (3.5%)	3 (3.1%)	0.857
Paced (%)	22 (4.2%)	21 (5%)	1 (1%)	0.07	67 (13%)	56 (13.4%)	11 (11.2%)	0.489	102 (20.5%)	102 (25.4%)	16 (16.5%)	0.274

BPM: beats per minute; HAFB: hemianterior fascicular block; HPFB: hemiposterior fascicular block; LBBB: left bundle branch block; RBBB: right bundle branch block; * nonstimulated QRS.

## Data Availability

The data presented in this study are available upon request from the corresponding author. The data are not publicly available due to French legislation on access to databases.

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
