# Peer review of "Post-Dilatation of New-Generation Self-Expandable Transcatheter Aortic Valves Does Not Increase Atrioventricular Conduction Abnormalities"

_diagnostics, 2023, doi:10.3390/diagnostics13030427_

Round 1

Reviewer 1 Report

This study reports that post-dilatation of the 2nd and 3rd generation Corevalve™ self-expandable TAVR valves does not increase the risk of developing conduction disturbances or permanent pacemaker implantation, and the use of the latest generation SE-TAVR considerably reduces peri-prosthetic regurgitation and the need for post-dilatation, hence is interesting and clinically significant. 

There are many grammatical/spelling errors to be corrected.

For example, ‘No specific guideline exist regarding balloon post-dilatation usually performed to reduce reduce peri-prosthetic leakage or prosthesis under-deployment associated with pour outcomes’ should be corrected as ‘No specific guideline exists regarding balloon post-dilatation usually performed to reduce peri-prosthetic leakage or prosthesis under-deployment associated with poor outcomes’.

Additionally, the references should be formatted.

Author Response

We thank the reviewer for his/her comment. Accordingly to his/her request we ask for an english editing.

References are now formated.

Reviewer 2 Report

Authors investigated and compared the effect of postdilatation among self-expandable Corevalve generations. Postdilatation of TAVR valves are required to ensure valve stability, and more importantly, to avoid paravalvular leak and later left ventricular volume overload, haemolysis. Major findings of the manuscript are that 2nd gen Evolut R valves require postdilatation more commonly than 3rd gen Evolut Pro, but this was not associated with more complications, such as pacemaker implantation. Noteworthy, it does not influence 1year mortality significantly.

Comments:

1) As mentioned above, postdilatation is required to avoid paravalvular leak. In some cases, real life scenarios dictate not to push the limit and considering cost-benefit, certain severity of residual paravalvular leak might be accepted. Authors should present the rate, severity and evolution of residual paravalvular leak after postdilatation comparing 2nd and 3rd gen Corevalve interventions.

2) Residual aortic regurgitation might also be an indicatior for quality of life, which could be indicated with many indicis (e.g. NYHA stage). Even though mortality data are disclosed, authors are required to present quality of life measures as well.

3) Reviewer is aware that the aim of the study was not to compare self-expandable and balloon-expandable TAVR valves. However, differences and comparison should be mentiones and a bit detailed in the Discussion section.

Author Response

Response to Reviewer 2.

  • As mentioned above, postdilatation is required to avoid paravalvular leak. In some cases, real life scenarios dictate not to push the limit and considering cost-benefit, certain severity of residual paravalvular leak might be accepted. Authors should present the rate, severity and evolution of residual paravalvular leak after postdilatation comparing 2nd and 3rd gen Corevalve interventions.

In order to clarify the results, we describe the set of postprocedural regurgitation, whether the valve was postdilated for malapposition or moderate/severe leak.

According to reviewer’s comment we now state in the Results chapter, Procedural characteristics section:

“We observed post-dilatation significantly less often in the Corevalve Evolut Pro group (Evolut R vs. Evolut Pro: 21.6 % vs. 12.1 %, p=0.024). Before post-dilatation, 77 patients had a moderate or severe regurgitation (nine due to malapposition): 66 (85.7%) patients with a second-generation Corevalve and 11 (14.3%) patients with a third-generation Corevalve. In total, 54 (51.4%) and 23 (22.1%) patients experienced moderate and severe regurgitation, with no difference between the valve types (p=0.532). After post-dilatation, 36 patients (34.6%) had a moderate leak and 3 (2.9%) a severe leak. No significant difference according to the valve type existed (see Supplementary Materials 3, p=0.378).“

And in supplementary materials :

Total

(n=104)

Corevalve Evolut R

 (n= 90)

Corevalve Evolut Pro

 (n=14)

p-value

Post-implantation regurgitation

0.532

None or mild

27 (26.0%)

24 (26.7%)

3 (21.4%)

Moderate

54 (51.9%)

44 (48.9%)

10 (71.4%)

Severe

23 (22.1%)

22 (24.4%)

1 (7.1%)

Post-dilatation regurgitation

None or mild

65 (62.5%)

54 (60.0.%)

11 (78.6%)

0.378

Moderate

36 (34.6%)

33 (36.7%)

3 (21.4%)

Severe

3 (2.9%)

3 (3.3%)

0 (0%)

Supplementary Materials 3. Post-implantation aortic regurgitation severity after SE-TAVR implantation

TAVR : transcatheter aortic valve replacement

  • Residual aortic regurgitation might also be an indication for quality of life, which could be indicated with many indicis (e.g. NYHA stage). Even though mortality data are disclosed, authors are required to present quality of life measures as well.

The follow-up of our cohort was performed at 1 year and focused on post-procedural death. The follow-up of the patients was not exclusively in our center. Among the 104 patients who underwent postdilatation, it was possible to obtain NYHA status for 80 patients.

According to reviewer’s comment we now state in the Result chapter, TAVR clinical outcomes section :

“No significant differences existed in the occurrence of complications per- and postprocedure between patients with and without post-dilatation. At 1 year, no difference in NYHA status existed (n=80, 2.0±0.6 vs. 2.0±0.6, p=0.405). The all-cause mortality at 30 days and 1 year were not significantly different between the groups (2.4%, p=0.302 at 30 days and 10.3%, p=0.507 at 1 year). Of the patients who died and received post-dilatation, six (75%) had moderate-to-severe leakage.”

  • Reviewer is aware that the aim of the study was not to compare self-expandable and balloon-expandable TAVR valves. However, differences and comparison should be mentiones and a bit detailed in the Discussion section.

According to reviewer’s comment we now state in the Discussion chapter, Study limitation section :

“We excluded balloon expandable valves from our cohort. Of the 362 patients implanted, only 1 had received post-dilatation. In the most recent literature, patients undergoing the post-dilatation of an expandable balloon or self-expanding valve shared similar clinical characteristics, with no mechanical complications or impact in terms of 1-year mortality. No description of per-procedure electrocardiogram changes is available for patients undergoing the post-dilatation of a self-expanding valve.“

Round 2

Reviewer 2 Report

I dont have any further comment.